# Myeloperoxidase Gene Deletion Causes Drastic Microbiome Shifts in Mice and Does Not Mitigate Dextran Sodium Sulphate-Induced Colitis

**DOI:** 10.3390/ijms25084258

**Published:** 2024-04-11

**Authors:** Patrick T. San Gabriel, Thomas R. O’Neil, Alice Au, Jian K. Tan, Gabriela V. Pinget, Yuyang Liu, Genevieve Fong, Jacqueline Ku, Elias Glaros, Laurence Macia, Paul K. Witting, Shane R. Thomas, Belal Chami

**Affiliations:** 1Charles Perkins Centre, School of Medical Sciences, Faculty of Medicine and Health, The University of Sydney, Sydney, NSW 2000, Australiapaul.witting@sydney.edu.au (P.K.W.); 2Rheumatology Department, Royal Prince Alfred Hospital, Camperdown, NSW 2050, Australia; 3Cardiometabolic Disease Research Group, Department of Pathology, School of Medical Sciences, Faculty of Medicine, University of New South Wales, Sydney, NSW 2052, Australiae.glaros@unsw.edu.au (E.G.);

**Keywords:** inflammatory bowel disease, colitis, microbiome, myeloperoxidase, ulcerative colitis, Crohn’s disease

## Abstract

Neutrophil-myeloperoxidase (MPO) is a heme-containing peroxidase which produces excess amounts of hypochlorous acid during inflammation. While pharmacological MPO inhibition mitigates all indices of experimental colitis, no studies have corroborated the role of MPO using knockout (KO) models. Therefore, we investigated MPO deficient mice in a murine model of colitis. Wild type (Wt) and MPO-deficient mice were treated with dextran sodium sulphate (DSS) in a chronic model of experimental colitis with three acute cycles of DSS-induced colitis over 63 days, emulating IBD relapse and remission cycles. Mice were immunologically profiled at the gut muscoa and the faecal microbiome was assessed via 16S rRNA amplicon sequencing. Contrary to previous pharmacological antagonist studies targeting MPO, MPO-deficient mice showed no protection from experimental colitis during cyclical DSS-challenge. We are the first to report drastic faecal microbiota shifts in MPO-deficient mice, showing a significantly different microbiome profile on Day 1 of treatment, with a similar shift and distinction on Day 29 (half-way point), via qualitative and quantitative descriptions of phylogenetic distances. Herein, we provide the first evidence of substantial microbiome shifts in MPO-deficiency, which may influence disease progression. Our findings have significant implications for the utility of MPO-KO mice in investigating disease models.

## 1. Introduction

The aetiology of inflammatory bowel diseases (IBD), including Crohn’s disease (CD) and ulcerative colitis (UC), is generally multifactorial, though it is primarily exacerbated via the dysfunctional interplay between the microbiome and the immune system [1,2]. Patients with IBD commonly exhibit characteristic recurrent leukocytic infiltration as a driving factor of inflammation in the early stages of disease pathogenesis. While neutrophil-derived myeloperoxidase (MPO) is critical for the catalytic formation of the microbicidal oxidant hypochlorous acid (HOCl), chronically elevated MPO levels and corresponding enzyme activity results in irreparable oxidative damage to host tissue [3,4].

MPO is a highly cationic dimeric heme-containing perxoxidase consisting of two heavy and light chains which uses H_2_O_2_ as a primary substrate. Respiratory burst is the primary cellular source of H_2_O_2_ utilised by MPO in immune cells of the myeloid lineage, including macrophages and neutrophils which both incidentally produce MPO. One of the main described reactions catalysed by MPO is the formation of the two-electron oxidant, hypochlorous acid (HOCl).

Dichotomous findings for the role of MPO in several disease settings are not uncommonly described. For example, pharmacological inhibition of MPO via AZM198 prevents plaque stability in a murine model of Tandem Stenosis, while the same MPO inhibitor stabilises atherosclerotic plaques in pre-existing atherosclerotic plaques [5] in various substrate inhibitors mitigating the course of disease. Paradoxically, MPO-deficient mice exhibited accelerated atherosclerosis [6].

MPO is believed to play a role in autoimmune encephalomyelitis, based on reports of MPO-laden foci plaques with increased inflammatory cell infiltration and demyelination [7]. However, the role of the role of MPO in autoimmune encephalomyelitis is also dichotomously described. In murine autoimmune encephalomyelitis, pharmacological inhibition of MPO via N-acetyl lysyltrosylcysteine amide, reduced demyelination and axonal injury, while restoring the blood-brain barrier integrity [8,9]. However, MPO-deficient (MPO KO) mice subjected to autoimmune encephalomyelitis showed significantly increased disease severity, with 90% of MPO deficient mice exhibiting complete hind limb paralysis, compared to just 33% of their Wt counterparts. 

Colonic MPO levels and associated neutrophil extracellular traps [10] are significantly increased and correlated to disease severity in patients with Crohn’s disease. Furthermore, pharmacological inhibition of MPO using the cyclic nitroxide, 4-methoxy-2,2,6,6-tetramethyl-piperidin-1-yl-oxyl (4-methoxy-TEMPO) as a competitive substrate [11] or the thioxanthene, AZD3241, a specific pharmacological inhibitor of MPO [12], all abrogated disease indices during experimental colitis. These outcomes are consistent with an earlier study using 4-hydroxy-2,2,6,6-tetramethyl-piperidin-1-yl-oxyl (TEMPOL) [13] that also protected the colon from experimental IBD. Collectively these data indicated that MPO is a viable therapeutic target in the setting of IBD, including for UC and CD. 

Herein, we examined the impact of genetic manipulation of MPO to study its role in a widely employed murine model of experimental colitis and attempted to describe the possible mechanisms that interplay with MPO-deficiency and the immune response during inflammation.

## 2. Results

### 2.1. MPO KO Mice Showed Significantly Poorer Clinical Outcomes after Acute Colitis Phases

During the first recovery phase, a significant loss in average body weight was detected in Wt-DSS and MPO KO DSS mice (Figure 1A,B). This outcome was corroborated via a lower disease activity index (DAI) in mice provided DSS in drinking water, (Figure 1C). Wt-DSS and MPO KO DSS groups yielded similar DAI scores at each phase of colitis induction in the chronic DSS model. Additionally, significant shortening of the colon was seen in the Wt-DSS and MPO KO DSS groups, relative to control counterparts but were not different between DSS-treated groups (Figure 1D). Overall, the absence of MPO did not affect DSS colitis clinical outcomes. Lastly, end-point indicators of colitis severity were increased in MPO KO DSS mice (*p* ≤ 0.001) compared with Wt-DSS mice, and both groups recorded significantly increased spleen weight to bodyweight ratios (*p* ≤ 0.001 and *p* < 0.5, respectively) relative to their control counterparts (Figure 1E).

### 2.2. MPO KO Mice Displayed Poor Colon Histopathology in DSS Colitis

Hallmark histopathological features of UC include colonic epithelial and crypt damage. Histopathological outcomes of colons from MPO KO DSS mice were largely equal or worsened when compared to the colons isolated from Wt-DSS mice. There was a significant increase in colonic epithelial damage (denoted by Q in Figure 2A–E), crypt damage (*p* < 0.05 and *p* ≤ 0.01, respectively; Appendix A) and partial goblet cell depletion (Appendix A) in both the Wt-DSS group (*p* < 0.05) and in MPO KO DSS mice relative to their respective controls (Appendix A).

In contrast, no differences were detected in goblet cell mucin production as determined via alcian blue staining between Wt-DSS and MPO KO DSS colons, indicating that there was no conferred protection from colitis in the genetically engineered mice (Appendix A). Fibrogenesis was also assessed as a marker of tissue injury and healing; both MPO KO DSS and Wt-DSS groups showed an increased trend in collagen deposition compared to non-challenged mice with no significant difference in collagen deposition between MPO KO DSS and Wt-DSS groups (Appendix A). Indicators of active inflammation included oedema and leukocyte infiltration (Figure 2A–E, denoted by R), which were increased in both Wt-DSS (*p* ≤ 0.01 and *p* ≤ 0.001, respectively) and MPO KO DSS groups (*p* ≤ 0.001) relative to their respective control (Appendix A). As anticipated, the histopathological score increased significantly in Wt-DSS (*p* ≤ 0.0001) and MPO KO DSS (*p* ≤ 0.0001) groups relative to their control counterparts, with no discernible difference between MPO KO DSS and Wt-DSS groups (Figure 2E and Appendix A). Overall, consistent with the clinical outcomes in Figure 1, the absence of MPO in MPO knockout mice did not affect colonic histopathology in DSS colitis, conferring that there was no colonic protection evident when compared to DSS-treated Wt mice.

### 2.3. MPO KO Immunophenotyping Pre- and Post-DSS Treatment 

Outcomes of experimental colitis are directly linked to colonic immune infiltration and activation [14,15,16], and as such, colonic leukocytes were evaluated via flow cytometry. Here, we determined a proportional enrichment of CD4^+^ T cells in both intraepithelial and lamina propria tissues from all DSS-colitis groups (Figure 3A,B) which was complemented with an increase in IL-17-producing cells (Figure 3C,D). These groups also experienced marked epithelial degradation of colon tissues (Appendix A). 

Th17 cells are distinguished by their functional cytokine IL-17 and in acute tissue damage, play a role in promoting tissue repair [17,18,19]. Notably, the overall landscape of inflammatory monocyte populations such as Ly6Chi (inflammatory) [20], neutrophils and IL-17 producing CD4^+^ T cells were unchanged in MPO KO DSS and Wt-DSS treated mice (Figure 3A–H).

### 2.4. The Gut Microbiome Was Substantially Altered in DSS-Challenged Wt and MPO KO Mice

The gut microbiota has been shown to contribute to colitis severity and to shift during disease progression. On Day 1, there was a significant shift in the gut microbiota composition, as assessed via 16S rRNA gene sequencing in both the Wt-DSS and MPO KO DSS groups (Figure 4). Thus, the composition of gut microbiota differed significantly between Wt and MPO KO mice indicating that absence of the immune protein MPO had an impact on gut homeostasis. Twenty-nine days post-treatment, both groups exhibited a shift in their gut microbiota with a distinct composition between the groups.

Furthermore, faecal bacterial microbiota in both Wt-DSS and MPO KO DSS mice was determined to be drastically distinct at Day 1, with a similar shift and distinction at Day 29, via qualitative (Figure 4A; Appendix A) and quantitative (Figure 4B) descriptions of their phylogenetic distances. Considering the richness and evenness of bacterial populations and corresponding ignorance of evolutionary divergence, a Bray-Curtis dissimilarity analysis also demonstrated distinct populations in each group (Figure 4C), with further visualisation of the highlighted difference depicted in a concurrent network analysis map (Figure 4D). Notably, these data confirmed that the microbiome was altered by MPO gene deletion in mice, which has implications for the baseline physiological homeostasis of engineered mice in comparison to their wild-type counterparts. Therefore, this outcome must be considered when making comparisons between engineered mice that lack key immune proteins and their WT counterparts.

Immunomodulatory effects of some bacterial genera have been well-described during colitis. We assessed the abundance of the top 10 relative genera which revealed notable differences in abundance of *Allobaculum*, *Lactobacillus* and *Eisenbergiella* across Wt and MPO KO and Day 1 and Day 29 time-points (Figure 5A–D). Bacterial species from the *Allobaculum* have been previously isolated in patients with IBD and identified as an IBD-associated genus owing to reports of exacerbating colitis [21,22]. MPO KO mice showed a significant increased abundance of the *Allobaculum* genus at Day 29, compared to all other groups and time-points (Figure 5B). Moreover, many bacterial species from the *Lactobacillus* genus are well-documented for their anti-inflammatory effects during colitis [23]. The relative abundance of the *Lactobacillus* genus was significantly higher in Wt mice at Day 1, compared to Day 29 and all time-points associated with MPO KO mice (Figure 5C). MPO KO mice showed no change in abundance of *Lactobacillus* at both Day 1 and Day 29, however the relative abundance of *Lactobacillus* was significantly lower than in Wt mice at Day 1. Interestingly, *Eisenbergiella* has no reported immunomodulatory effects in the gastrointestinal tract. We observed no differences in the relative abundance of *Eisenbergiella* in MPO KO mice at both Day 1 and Day 29, nor were there any significant differences in Wt counterparts at any time-point, hence no relative abundance shifts were noted from Day 1 to Day 29 in any mice genotype (Figure 5D).

## 3. Discussion

In contrast to pharmacological inhibition studies targeting MPO, herein MPO KO mice challenged with DSS revealed poor clinical and histopathological outcomes, with no observed protection from acute experimental colitis. In addition, we observed no differences in the colonic immune environment upon enumeration of major immune cell subtypes within the colonic environment in MPO KO and corresponding Wt mice. Interestingly, we were the first group to report significant and persistent shifts in phylogenetic distances in the faecal microbiome of MPO KO mice compared to Wt mice monitored over a 29-day period. Our study attempted to delineate the role of MPO during experimental colitis via the MPO gene deletion approach, as opposed to our previous attempts using pharmacological interventions. While we have previously reported that MPO inhibition via pharmacological intervention using either cyclic nitroxides (4-methoxy-TEMPO) or 2-thioxanthines (AZD3421) ameliorates DSS-induced colitis, we now report that MPO KO mice showed poor colitis outcomes during acute experimental colitis [11,12].

The dichotomy in delineating the role of MPO via pharmacological blockage and gene deletion has been reported previously in other inflammatory models. For example, while two studies found that pharmacological inhibition and MPO deletion improved atherosclerosis in mice models [24,25], another found that MPO depletion worsened the atherosclerotic condition [26].

A similar contradiction was observed between independent research groups in an identical model of myelin oligodendrocyte glycoprotein (MOG) antibody-associated encephalomyelitis/encephalitis. In this experimental pathology analysis, MPO KO mice were shown to be more susceptible to the MOG-encephalomyelitis model and showed poor clinical and biochemical outcomes, whereas pharmacological inhibition of MPO using the N-acetyl lysyltyrosylcysteine amide MPO inhibitor restored blood-brain barrier function and attenuated disease progression in the same encephalomyelitis model [26].

However, not all MPO genetic deletion models reported a dichotomous character. In a murine model of Alzheimer’s disease, MPO deficiency resulted in significantly reduced levels of inflammatory mediators in the hippocampus and this was correlated with superior performance in spatial learning and memory [27]. 

MPO, via the production of HOCl and other oxidants, is a bactericide. We now provide evidence of substantial microbiome shifts in MPO KO mice that persisted over a 29-day period. A recent study identified the gut microbiota as a key driver of DSS-induced colitis variability [28]. Specifically, the relative abundance at the genus level of *Duncaniella* and *Alistipes* was positively correlated to worsened disease outcomes after DSS treatment. We showed an increase in the relative genus abundance of *Alistipes* in MPO KO mice compared to Wt mice on Day 29 but not on Day 1. This observation suggested that an increase in this bacterial population may be implicated in disease progression in the affected colon.

In addition, the abundance of the genus *Lactobacillus* in MPO KO mice was significantly lower on both Day 1 and Day 29, compared to Wt mice on Day 1. Numerous studies have ascribed an anti-inflammatory role for many *Lactobacillus* species, particularly in the context of colitis. For example, *Lactobacillus brevis* is a well-described probiotic with anti-inflammatory gut activity and has previously been shown to alleviate DSS-induced colitis while also influencing the serum metabolome, hence also providing systemic effects [29]. Similarly, *Lactobacillus rhamnosus* and *Lactobacillus salivarius* protected mice from TNBS-induced colitis via switching DC differentiation towards tolerogenic phenotypes defined by their inability to produce cytokines, chemokines or co-stimulatory molecules for T cell activation [30]. In UC patients, the intestinal abundance of *Lactobacillus* was positively correlated to reduced clinical symptoms [31]. 

Some species of the *Allobaculum* genus are associated with the worst colitis outcomes, specifically *Allobaculum mucilyticum* and *Allobaculum fili* [21]. The mechanisms by which some *Allobaclum* species exacerbate colitis are unclear, however, *Allobaculum mucolyticum* has been shown to degrade human intestinal mucin [22] and disruption to the mucin-barrier is associated with IBD. The relative abundance of the *Allobaculum* genus was significantly increased in MPO KO mice at Day 29 and this could explain the susceptibility of colitis in MPO KO mice. 

*Eisenbergiella* is not known to be associated with IBD in either humans or experimental models. We found no shifts in relative abundance over time in both Wt and MPO KO mice, however we observed a trending increase in *Eisenbergiella* in MPO KO mice, compared to their Wt counterparts. It is unclear if this played a role in our model or has implications for colitis. 

Our limitation was that our measurement was restricted to the genus level and we did not examine the species of bacteria across groups.

The observed changes to the gut microbiome and the relative abundance of certain genera following MPO deletion may have influenced the outcome of DSS-induced colitis when compared to the pharmacological blockage of MPO. This may explain the dichotomy observed in this study when compared to our previous studies of pharmacological inhibition of MPO and improved clinical outcomes in DSS-induced colitis.

It is possible that MPO gene deletion may impact the immunological gut milieu and this could have altered the clinical outcome of experimental colitis. However, our immunological phenotyping of colonic lamina propria leukocytes and intra-epithelial lymphocytes showed no differences compared to Wt controls, suggesting MPO gene deletion did not significantly impact the distribution and major immunological constituents within the gut. However, as we did not assess the gut cytokine milieu or lymphocyte subtypes, this represents a limitation to our study.

During the DSS challenge, neutrophil migration to the inflamed colitis in MPO KO mice significantly increased relative to control counterparts. A recent study suggested that released MPO attenuates further neutrophil migration and may explain the enrichment of neutrophils in DSS-treated MPO KO mice [32].

The strength of our study was the evaluation of the microbiome at two time-points during chronic colitis. This revealed distinct phylogenetic polarities that are largely maintained with time and differences in the relative abundance of some IBD-associated genera in Wt and MPO KO mice which may influence the course of colitis. 

## 4. Materials and Methods

All materials were purchased from Sigma-Aldrich (North Ryde, NSW 1670, Australia) unless otherwise specified and were of the highest quality available. Colitis grade DSS was sourced from MP Biomedicals, Santa Ana, CA, USA (36,000–50,000 Da). Antibodies were sourced from Biolegend, San Diego, CA, USA which included CD4 (GK1.5), CD45 (30-F11), IL-17 (TC11-18H10.1), TCRgd (15-5711-82), Ly6C (HK1.4) and Ly6G (1A8). Faecal DNA isolation kits were purchased from Bioline, London, UK, BIO-52082) and the Qubit™ dsDNA BR assay kit was sourced from ThermoFisher Scientific, Waltham, MA, USA (Q32850).

### 4.1. Mice and Ethics

Female MPO gene knockout (MPO KO) C57BL/6J mice and corresponding C57BL/6J wild type (Wt) mice were obtained from Australian BioResources, Moss Vale, NSW, Australia. The MPO gene knockout was confirmed via RT-PCR melt curve analysis (forward primer: GCAGTTCAGGGTTGTGTGGTGTA, reverse primer: GGGCTAGAGAGGACCTAGGACTC) (Appendix A). Female mice were chosen in this study as male mice develop more significant and aggressive disease during DSS-induce colitis. Mice were aged-matched (7 weeks on arrival) and weight-matched (16–17 g ± 0.5). Subsequently, mice were housed in the Charles Perkins Centre Animal House (University of Sydney, Australia) in accordance with institutional animal ethics guidelines and with ad libitum access to food and water. Mice were acclimatized for 1 week prior to the commencement of experiments. Mice were then randomly assigned to their genotype-specific counterparts (wild-type mice, Wt; MPO deficient mice, MPO^−/−^) and received either water (CTRL) or DSS (DSS) in their drinking water.

### 4.2. Experimental Colitis

Mice were randomly assigned to control and treatment groups prior to the murine experimental colitis model, which involved 3 acute cycles of orally administered DSS in mice drinking water (ad libitum), followed by periods of recovery emulating human UC disease relapse and remission [33] (Figure 6). Briefly, colitis was first induced with 2.5% (*w*/*v*) DSS (first cycle) and 2% (*w*/*v*) DSS thereafter (second and third cycles) for 7 days followed by 14-day recovery periods with cycling occurring over a total of 21 days. Individual body weight, faecal consistency and rectal bleeding were assessed as previously described [34].

### 4.3. Tissue Sample Collection and Preparation

Anaesthesia was induced with 3% (*v*/*v*) isoflurane, followed by cardiac puncture to collect blood and subsequent cervical dislocation before organ harvest. Excised colons were flushed with ice-cold PBS and colonic length was measured before dissecting the tissue longitudinally into two equal halves. Colons were fixed overnight with cold 70% ethanol and later processed for histology and histochemistry. 

### 4.4. Histopathological Staining and Scoring

Following tissue processing, fixed colon tissue was stained with haematoxylin and eosin and alcian blue/acetic safranin [10], as well as picrosirius red [35]. Imaging and histopathological scoring were conducted using the Fiji distribution of ImageJ software (version 1.53f51). Pathological scores were calculated based on histopathological parameters detailed in Appendix A and were conducted in a blinded fashion by experienced histologists.

### 4.5. Flow Cytometry Analysis

Intraepithelial cells were isolated by incubating minced colon at 37 °C for 40 min in Hanks’ balanced salt solution with 5 mM EDTA, 5% *v*/*v* foetal bovine serum (FBS) and 15 mM HEPES. Lamina propria cells were isolated by incubating at 37 °C for 60 min in RPMI with 2 mg/mL collagenase type IV, 10% (*v*/*v*) FBS and 15 mM HEPES. Leukocytes were enriched via percoll gradient centrifugation. Single cell suspensions were stained using the antibodies listed in Appendix A and analysed on an LSR II 5L and FlowJo software (version 10.0).

### 4.6. DNA Amplicon Sequencing Targeting the 16S rRNA Gene 

Faeces were collected on Days 1 and 29 under sterile conditions. Faecal DNA was extracted and purified according to the manufacturer’s instructions (Bioline; BIO-52082). DNA was quantified using a Qubit™ dsDNA BR assay kit. DNA amplifiability was demonstrated via PCR before submission of samples to Ramaciotti Centre for Genomics (University of New South Wales, Australia) for 16S rRNA gene faecal microbial amplicon sequencing (V1–V3 region) and subsequent analysis via DADA2 methodology [36]. 

### 4.7. Statistical Analyses

Statistical analysis was performed on GraphPad Prism (version 7.02). All data sets were tested and judged to be normally distributed via Kolmogorov–Smirnov and Shapiro–Wilk tests. Between-group comparisons were then made using one-way analysis of variance with Tukey’s correction. All data is presented as mean ± standard deviation unless otherwise specified, with significance established if *p* < 0.05.

## 5. Conclusions

Our previous studies using pharmacological inhibition of MPO in Wt mice improved overall clinical and colon histopathological outcomes during challenge with acute DSS-induced colitis. However, interestingly we observed no protective effects in MPO KO mice when challenged with DSS in this study. MPO KO mice showed significant faecal microbiota divergence via a Bray-Curtis dissimilarity analysis to Wt mice on both Day 1 and Day 29 of our study. These microbial shifts likely contributed to the negative outcomes of MPO KO mice observed in this study, compared to protective effects with pharmacological MPO inhibition during DSS challenge. Our findings point to a dichotomy in the role of MPO during acute colitis and have significant implications for interpreting other disease models of inflammation where MPO plays a pathological role. Considering our findings described herein, alterations to the microbiome should be evaluated when utilising MPO KO mice in disease models, particularly when outcomes differ from pharmacological MPO inhibition. Whether these outcomes are also relevant to other mice where immune proteins are genetically eliminated remains to be established.

## Figures and Tables

**Figure 1 ijms-25-04258-f001:**
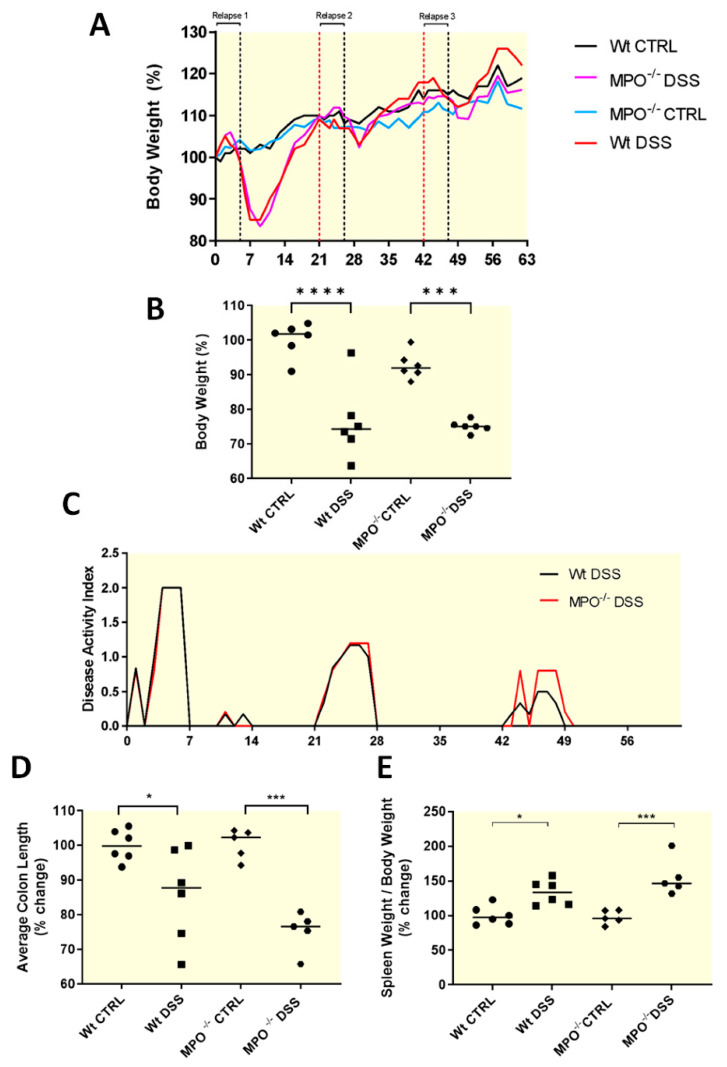
Assessment of clinical parameters in DSS-induced chronic colitis. 6–7-week-old female C57BL/6 mice were subjected to 2–2.5% *w*/*v* DSS-induced colitis during three iterations of seven-day disease relapse phases over 63 days. (**A**) Percentage body weight over the entire duration of the 63-day model. (**B**) Average body weight loss on Day 9. (**C**) Disease activity index was measured by enumerating daily rectal bleeding (0 = absent, 1 = slight, 2 = moderate, 3 = extreme) and stool consistency (0 = normal, 1 = loose, 2 = perineal soiling, 3 = extreme perineal soiling) scores. (**D**) Colons were resected between the ileocaecal junction and anal verge post-sacrifice at model endpoint as a marker of colitis severity. (**E**) Spleens were resected post-sacrifice at model endpoint and weighed (weight-to-bodyweight ratio) to determine the degree of splenomegaly as a feature of colitis severity. CTRL = control. DSS = dextran sodium sulphate. MPO KO = myeloperoxidase gene deficient. n = 6, all groups. Data shown as mean ± SD. * *p* < 0.05, *** *p* ≤ 0.001, **** *p*≤ 0.0001.

**Figure 2 ijms-25-04258-f002:**
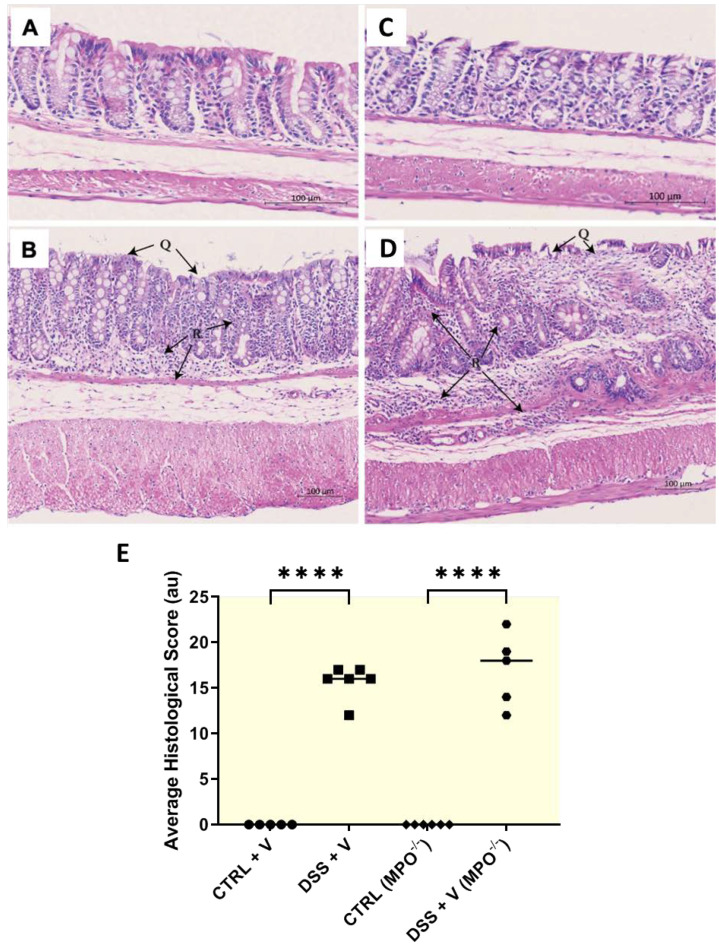
Representative colon sections stained with haematoxylin and eosin for histopathological scoring after the 63-day chronic colitis model. (**A**) Wt control, (**B**) Wt-DSS, (**C**) MPO KO control, (**D**) MPO KO DSS. Extensive epithelial damage (Q) and leukocyte infiltration are annotated in the representative images. (**E**) Enumeration of epithelial and crypt damage, partial goblet cell depletion, average oedema and leukocyte infiltration scores yielded an average histological score for each experimental group. Wt = wild type. MPO KO = myeloperoxidase gene deficient. DSS = dextran sodium sulphate. n = 6, all groups. **** *p* ≤ 0.0001.

**Figure 3 ijms-25-04258-f003:**
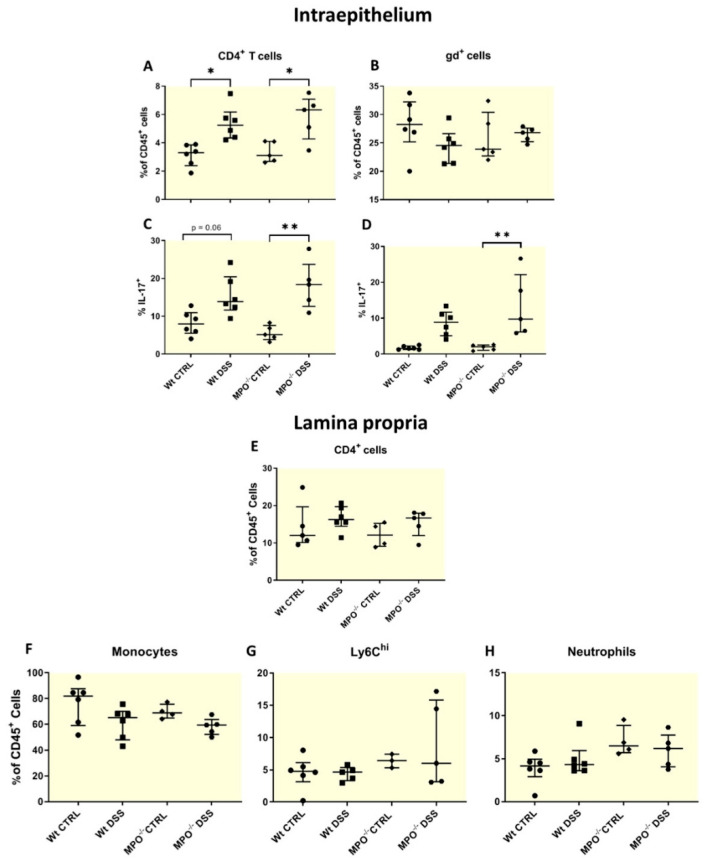
Immune cell profiling of mice colon via FACS. Percentage of (**A**) intraepithelial CD45^+^ leukocytes that are CD4^+^ T cells and (**B**) gd-T cells and the percentage of (**C**) IL-17^+^CD4^+^ T cells and (**D**) gd-T cells. Percentage of lamina propria CD45^+^ leukocytes that are (**E**) total CD4^+^ T cells and those that are IL-17^+^CD4^+^ T cells, (**F**) monocytes, (**G**) Ly6Chi inflammatory macrophages and (**H**) neutrophils. CTRL = control. DSS = dextran sodium sulphate. MPO KO = myeloperoxidase gene deficient. CD4 = cluster of differentiation 4. CD45 = cluster of differentiation 45. Gd-T cells = gamma delta T cells. IL-17 = interleukin 17. Ly6Chi = lymphocyte antigen 6 complex high. Data shown as median with interquartile range. n = 6 Wt CTRL and Wt DSS, n = 5 MPO^−/−^ CTRL and MPO^−/−^ DSS. * *p* ≤ 0.05, ** *p* ≤ 0.001.

**Figure 4 ijms-25-04258-f004:**
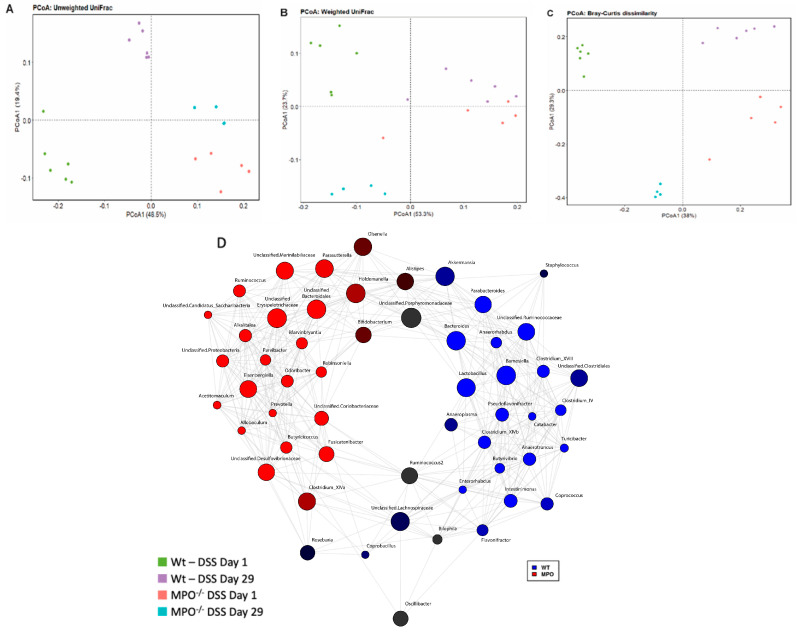
Beta diversity data from 16S rRNA gene microbial amplicon sequencing of chronic DSS colitis stool sample. (**A**) Principal coordinates analysis of unweighted UniFrac. (**B**) Principal coordinates analysis of weighted UniFrac. (**C**) Principal coordinates analysis of Bray-Curtis dissimilarity. (**D**) Co-occurrence network of bacterial genera within stool samples from wild type and MPO gene deficient mice on Day 29 of chronic DSS colitis. Lines connecting nodes represent the association between linked nodes. Greater intensities of red and blue correspond to higher observed amounts of the specific bacterial genus within the corresponding mouse genotype (Pearson correlation). n = 6 Wt—DSS Day 1, n = 6 Wt—DSS Day 29, n = 5 MPO^−/−^ DSS Day 1, n = 4 MPO^−/−^ DSS Day 29.

**Figure 5 ijms-25-04258-f005:**
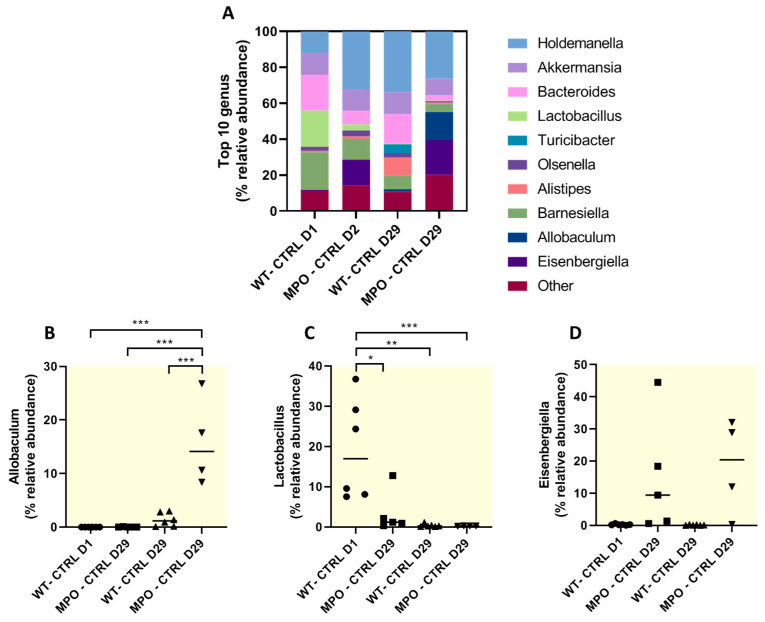
Analysis of the top 10 genera revealed from 16S rRNA gene microbial amplicon sequencing of Wt and MPO KO stool sample. (**A**) Representation of the top 10 genera and relative genus abundance of (**B**) *Allobaculum*, (**C**) *Lactobacillus* and (**D**) *Eisenbergiella* in Wt and MPO^−/−^ at Day 1 and Day 29. n = 6 Wt—DSS Day 1, n = 6 Wt—DSS Day 29, n = 5 MPO^−/−^ DSS Day 1, n = 4 MPO^−/−^ DSS Day 29. * *p* ≤ 0.05, ** *p* ≤ 0.001 & *** *p* ≤ 0.0001.

**Figure 6 ijms-25-04258-f006:**
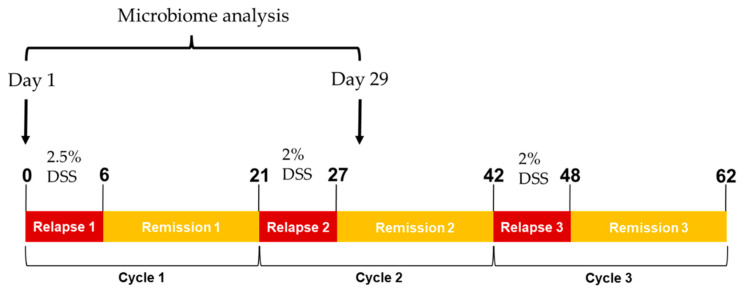
In vivo study design of DSS-induced chronic colitis.

## Data Availability

The data that support the findings of this study are available from the corresponding author upon reasonable request. Omic data pertaining to this study was deposited in the European Nucleotide Archive database under the accession code PRJEB66447.

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
