# Peer review of "Myeloperoxidase Gene Deletion Causes Drastic Microbiome Shifts in Mice and Does Not Mitigate Dextran Sodium Sulphate-Induced Colitis"

_ijms, 2024, doi:10.3390/ijms25084258_

Round 1

Reviewer 1 Report

Comments and Suggestions for Authors

The manuscript provided by San Gabriel and colleagues describes the effects of genetically induced loss of MPO in regard to pharmacologically evoked colitis in mice. The manuscript per se is well written and I especially appreciate the will to publish "negative" results. On first sight it might have been disappointing to observe no ameliorative effect of the loss of enzyme; however, it is important to describe this discrepancy between inhibiting the enzyme by small molecules and the full-blown genetic model. Besides some minor points, I have two major points of criticism that need to be addressed by the authors:

1) when describing lack of effect, it is - at least in my opinion - tremendously important to proof the presence of the genetic enzyme deletion. I would not suggest to only cite other publications in this regard but would see it as mandatory to demonstrate loss of MPO in neutrophils and other myeloid cells within the mice that were used in this study. I don't want to blame someone - however, mistakes may occur and as ko and wt-DSS mice were indistinguishable in regard to their reaction to challenge, it is needed to confirm that ko was really present. A simple enzymatic activity assay from isolated cells or a Western blot will do. Moreover, I would love to see a more detailed description on the ko model and also some references that describe the model in other contexts.

2) Description of the microbiome changes probably elicited by the genetic MPO ko is rather poor. There needs to be a better descritpion which genera change and if this gets significant etc.

Besides these important points, I have mainly small issues to clarify:

Wording is at some points a bit irritating. I would not suggest to describe MPO as a microbizide (line 14). The products synthesized by this enzyme might be called this term.

line 17 "...the utility of MPO deficient mice in a murine model.." This sentence needs to be rephrased. 

16SrRNA sequencing...it was a DNA sequencing attempt

What was the rationale of choosing day 1 and 29? Moreover, as the reader has not been presented the treatment schedule in the abstract, it might be better to use relative terms here. Such as "at the beginning of the treatment and after half of the time of treatment period".

line 42 "...dimeric heme-containing tetrameric"...this needs to be rephrased.

Please correct H2O2.

"the formation of plaque instability" (line 50): do the authors mean the induction of plaque instability?

line 55 and 61: language correction

End of the introduction: the two last paragraphs somehow are repetitive. I would suggest to shorten accordingly.

What was the rationale of only using female mice?

n numbers need to be indicated in the figure legends.

p<0.05 etc. instead of p<= 0.05

If the authors find a goblet cell depletion, how can they then obtain no differences in mucin production (Suppl. Fig. 1C and 2A-E)?

line 120 : Figure 2A-E instead of 1?

line 148 "monocytes and neutrophils were proportionally enriched" I cannot see this within Figure 3F and H. On the contrary, these cell types even seem to be lowered upon DSS treatment or stay the same. However, nothing statistically significant is indicated here.

Lactobacillus genus does not mean L. brevis. The authors themselves point at the limitation of their study in this regard. However, I would additionally suggest to add also an example of a non-beneficial Lactobacillus to give a balanced view.

How were the mice allocated to control and treatment? How was DSS administered? Did control mice receive a comparable treatment with solvent? Information on used material is poor. I would ask the authors to add the respective vendors at least. 

line 275 ##?

line 268: what kind of biochemical analyses were performed?

Maybe it would be valuable to also add Alzheimer's disease as an example of MPO-related immune suppression. There is one report that describes efficiency of the genetic deletion model, so it seems that not in all cases MPO exerts this dichotomous character (Volkman et al., Front Neurosci 2019). For a pharmacologic. intervention (in rats): Wang et al., Curr Med Sci 2022.

Comments on the Quality of English Language

There are some minor things to be fixed language-wise.

Author Response

Reviewer 1

The manuscript provided by San Gabriel and colleagues describes the effects of genetically induced loss of MPO in regard to pharmacologically evoked colitis in mice. The manuscript per se is well written and I especially appreciate the will to publish "negative" results. On first sight it might have been disappointing to observe no ameliorative effect of the loss of enzyme; however, it is important to describe this discrepancy between inhibiting the enzyme by small molecules and the full-blown genetic model. Besides some minor points, I have two major points of criticism that need to be addressed by the authors:

1) when describing lack of effect, it is - at least in my opinion - tremendously important to proof the presence of the genetic enzyme deletion. I would not suggest to only cite other publications in this regard but would see it as mandatory to demonstrate loss of MPO in neutrophils and other myeloid cells within the mice that were used in this study. I don't want to blame someone - however, mistakes may occur and as ko and wt-DSS mice were indistinguishable in regard to their reaction to challenge, it is needed to confirm that ko was really present. A simple enzymatic activity assay from isolated cells or a Western blot will do. Moreover, I would love to see a more detailed description on the ko model and also some references that describe the model in other contexts.

Thank you for this suggestion. We now have included in our Supplementary figures (Supplementary Figure 5) melt curve analysis for the MPO gene. We compared Wt to MPO KO mice and included a heterogeneous control from randomly selected mice in their designated genotypes. We also included this now in our method (line 337).

2) Description of the microbiome changes probably elicited by the genetic MPO ko is rather poor. There needs to be a better descritpion which genera change and if this gets significant etc.

Thank you for point this out. We have reapproached our analysis to better by extracting the top 10 genera changes and performing analysis on 3 of the most prominent changes in relative abundance between Wt mice and MPO KO mice. This is now Figure 5 with associated figure legend and results and discussion expanded (line 189-205; 218-222; 269-292)

We also added a new figure to better support this discussion (i.e Figure 5). We broke down the top 10 genus and analysed the relative abundance of Allobaculium, Lactobacillus and Eisenbergiella. We should that

Besides these important points, I have mainly small issues to clarify:

Wording is at some points a bit irritating. I would not suggest to describe MPO as a microbizide (line 14). The products synthesized by this enzyme might be called this term.

We agree with the reviewer that HOCl and other oxidative products derived from MPO are microbicidal and not directly MPO per se. As such, we have now corrected all instances of this in our paper (page 1, line 13 & page 1, line 40-41).

line 17 "...the utility of MPO deficient mice in a murine model.." This sentence needs to be rephrased. 

Thank you for pointing out this misphrased sentence. We have now made the appropriate amendments (page 1, line 16 )

16SrRNA sequencing...it was a DNA sequencing attempt

Thank you for pointing this out. We have clarified this further by changing ‘16S rRNA sequencing’ to ‘16S rRNA gene sequencing’. (page 5, line 171 & page 6, line 188 & page 9 & line 330).

We also changed the subheading to ‘DNA amplicon sequencing targeting the 16S rRNA gene’ to remove the ambiguity. (page 9, line 325).

We wanted to examine if the microbiome would converge with time when the mice were housed in the same facility and general environment. It was previously described that micribiome normalization could occur if as little as a month. Our results indicted that faecal bacterial microbiota in both Wt-DSS and MPO KO DSS mice was determined to be drastically distinct at day 1, with a similar shift and distinction at day 29. Thus they did not converge and maintained distinct polarities at day 1 and at least until our half way point of our study.

What was the rationale of choosing day 1 and 29? Moreover, as the reader has not been presented the treatment schedule in the abstract, it might be better to use relative terms here. Such as "at the beginning of the treatment and after half of the time of treatment period".

We have now ascribed day 1 as the ‘beginning of treatment’ and day 29 and the half-way point in our abstract (page 1, line 24-25). We assessed if the microbiomes of our Wt and MPO KO mice converged following 30 days of housing in the same environment following their arrival to our facility. Thus, we measured the microbiome at day 1 as a baseline measurement to compare at the 1 month mark. We found that although the respective microbiomes shifted from day 1 to day 30, they did not converged and maintained distinct polarities in respect to each genotype.

line 42 "...dimeric heme-containing tetrameric"...this needs to be rephrased.

We simplified this sentence to read ‘MPO is a highly cationic dimeric heme-containing perxoxidase consisting of two heavy and light chains which uses H2O2 as a primary substrate.’ (page 2, line 45-46).

Please correct H2O2.

Thank you for bringing this typographical error to our attention is now corrected (page 2, line 46 & 47).

"the formation of plaque instability" (line 50): do the authors mean the induction of plaque instability?

We have removed the ambiguity of this sentence and it now reads ‘For example, pharmacological inhibition of MPO via AZM198 prevents plaque stability in a murine model of Tandem Stenosis, while the same MPO inhibitor stabilises atherosclerotic plaques in pre-existing atherosclerotic plaques [5] in various substrate inhibitors mitigates the course of disease’ (page 2, line 54)

line 55 and 61: language correction

Both instances have now been corrected.

Line 55 ‘MPO inhibitor stabilises atherosclerotic plaques in pre-existing atherosclerotic’

Line 61 ‘However, the role of MPO in autoimmune encephalomyelitis is also dichotomously described’

End of the introduction: the two last paragraphs somehow are repetitive. I would suggest to shorten accordingly.

We have now shortened the last two paragraphs as to render the manuscript more succinct.

‘Colonic MPO levels and associated neutrophil extracellular traps [10] are signifi-cantly increased and correlated to disease severity in patients with Crohn’s disease.  Furthermore, pharmacological inhibition of MPO using the cyclic nitroxide, 4-methoxy-2,2,6,6-tetramethyl-piperidin-1-yl-oxyl (4-methoxy-TEMPO) as a competitive substrate [11] or the thioxanthene, AZD3241 – a specific pharmacological inhibitor of MPO [12], all abrogated disease indices during experimental colitis. These outcomes are consistent with an earlier study using 4-hydroxy-2,2,6,6-tetramethyl-piperidin-1-yl-oxyl (TEMPOL) [13] that also protected the colon from experimental IBD.  Collectively these data indicate that MPO is a viable therapeutic target in the setting of IBD, include UC and CD. Herein, we examined the impact of genetic manipulation of MPO to study its role in a widely employed murine model of experimental colitis and attempt to describe the possible mechanisms that interplay with MPO-deficiency and the immune response during inflammation.’ (page 2, line 70-86).

What was the rationale of only using female mice?

The dextran sodium sulphate (DSS) model is performed predominately on female C57BL/6 mice as they exhibit a slightly milder course of colitis, compared to male mice which typically do not survive day 5 post-DSS challenge. This enables the model to progress further and renders it more suitable for investigations. Moreover, our previous studies using pharmacological inhibition of MPO in DSS-induced colitis were performed in female mice and in order to draw appropriate comparisons, it was important to select female mice which have a similar course of colitis. We have now included an explanation in our methods section for the justification of female mice usage in our study.

‘Female mice were chosen in this study as male mice develop more significant and aggres-sive disease during DSS-induce colitis’ (page 7, line 267-269).

n numbers need to be indicated in the figure legends.

We thank the review for this important detail. We have now included the n numbers in each figure legend.

p<0.05 etc. instead of p<= 0.05

We have removed all <= references when relating to a p value of 0.05.

If the authors find a goblet cell depletion, how can they then obtain no differences in mucin production (Suppl. Fig. 1C and 2A-E)?

We observed a partial goblet cell depletion, which is typical in a DSS model of colitis and is linked to the severity of colitis (1). The partial goblet depletion was observed in both Wt mice and MPO KO mice when treated with DSS, with no differences between these two groups indicating a similar degree of colitis severity (page 3, line 122-124).

We have made this more clear but modifying the text slightly to reflect ‘partial’ goblet depletion (as opposed total goblet depletion) (page 3, line 119).

  1. Kaur, K., Saxena, A., Larsen, B., Truman, S., Biyani, N., Fletcher, E., Baliga, M.S., Ponemone, V., Hegde, S., Chanda, A. and Fayad, R. (2015). Mucus mediated protection against acute colitis in adiponectin deficient mice. Journal of Inflammation, 12(1). doi:https://doi.org/10.1186/s12950-015-0079-y.

line 120 : Figure 2A-E instead of 1?

Thank you for pointing out the discrepancy. We have changed the in-text reference to Figure 2A-E. (page 3, line 121).

line 148 "monocytes and neutrophils were proportionally enriched" I cannot see this within Figure 3F and H. On the contrary, these cell types even seem to be lowered upon DSS treatment or stay the same. However, nothing statistically significant is indicated here.

Thank you for this observation. We agree with the reviewer that there is no significant difference in monocyte and neutrophil populations and thus we have completed removed this sentence and instead have modified the sentence below.

‘Notably, the overall landscape of inflammatory monocyte populations such as Ly6Chi (inflammatory) (Rose et al., 2012), neutrophils and IL-17 producing CD4+ T cells were unchanged in MPO KO DSS and Wt-DSS treated mice (Figure 3A-H).’ (page 4, line 155-158).

Lactobacillus genus does not mean L. brevis. The authors themselves point at the limitation of their study in this regard. However, I would additionally suggest to add also an example of a non-beneficial Lactobacillus to give a balanced view.

We thank the reviewer for pointing out the ambiguity of this sentence. We have since made this clearer to the reader.

‘In addition, changes in levels and distribution of the genus Lactobacillus in MPO KO mice were also noted on both Day 1 and Day 29, whereas Wt mice show a relatively high abundance of genus Lactobacillus at both corresponding time points. Most species of lac-tobacillus are thought to have an anti-inflammatory role during colitis. For example, Llactobacillus brevis is a well-described probiotic with anti-inflammatory gut activity and has previously been shown to alleviate DSS-induced colitis while also influencing the se-rum metabolome, hence also providing systemic effects [18]. Similarly, lactobacillus rhamnosus and Lactobacillus salivarius protected mice from TNBS-induced colitis via switching  DC differentiation towards tolerogenic phenotypes defined by their inability neither to produce cytokines, chemokines or co-stimulatory molecules for T cell activation (10.1371/journal.pone.0000313). Our limitation is that our measurement was restricted to the genus level, and we did not examine the species of bacteria across groups.’ (page 7, line 240-251).

In regards to the addition of non-beneficial Lactobacillus in the setting of IBD or experimental colitis, we could not find references to support this, though we noted some references supporting proinflammatory cytokine product in other mucosa systems (i.e. vaginal) and some in vitro experiments. However, we felt this was outside of the scope of our experiments and decided to report only those references that were directly pertinent to colitis.  

How were the mice allocated to control and treatment? How was DSS administered? Did control mice receive a comparable treatment with solvent? Information on used material is poor. I would ask the authors to add the respective vendors at least. 

The mice were allocated to control and treatment group at random. Further, DSS was administered orally in drinking water, and we have now reflected this more clearly in the methods section.

‘Mice were randomly assigned to control and treatment groups prior to the murine experimental colitis model, which involved 3 acute cycles of orally administered DSS in mice drinking water (ad libitum), followed by periods of recovery emulating human UC disease relapse and remission (Wirtz et al., 2007).’ (page 8, line 277-278).

Admittedly, the Materials section of the manuscript is underwhelming. We have now included more information for the source of our raw materials, antibodies (clone information and source) as well as the source of our faecal DNA isolation kits

‘All materials were purchased from Sigma–Aldrich (North Ryde, NSW 1670, Australia) unless otherwise specified and were of the highest quality available.. Colitis grade DSS was sourced from MP Biomedicals (36,000 – 50,000 Da). Antibodies were sourced from Biolegend which included CD4 (GK1.5), CD45 (30-F11), IL-17 (TC11-18H10.1), TCRgd (15-5711-82), Ly6C (HK1.4) and Ly6G (1A8). Faecal DNA isolation kits were purchased from Bioline, BIO-52082) and Qubit™ dsDNA BR Assay Kit was sourced from Ther-moFisher Scientific (Q32850).’

line 275 ##?

The ‘##’ is now removed.

line 268: what kind of biochemical analyses were performed?

Line 268 refers to the Methods subheading ‘4.3. Tissue Sample Collection & Preparation’. We removed the sentence ‘The second section of colon was frozen in liquid nitrogen and stored at -80°C until re-quired for biochemistry. Spleens were resected and weighed before being frozen and stored in a similar manner’ as in this study, we did not utilize these organs for analysis.

Maybe it would be valuable to also add Alzheimer's disease as an example of MPO-related immune suppression. There is one report that describes efficiency of the genetic deletion model, so it seems that not in all cases MPO exerts this dichotomous character (Volkman et al., Front Neurosci 2019). For a pharmacologic. intervention (in rats): Wang et al., Curr Med Sci 2022.

We thank the reviewer for bringing this manuscript to our attention. In the interest of a well-balanced discussion, we have began a draft in the discussion for findings related to Volkman et al, Front NeuroSci 2019.

‘However, not all MPO genetic deletion models of report a dichotomous character. In a murine model of Alzheimer’s disease, MPO deficiency resulted in significantly reduced levels of inflammatory mediators in the hippocampus and this was correlated with supe-rior performance in spatial learning and memory (10.3389/fnins.2019.00990).’ (page 7; line 224-227).

However, after an exhaustive search we could not find the reference associated with Want et al., Curr Med Sci 2022. Moreover, we could not find references to support myeloperoxidase inhibitors used in AD models (except in the instance of MPO inhibitors used in PD models).

Could the reviewer kindly provide us with he DOI to help locate the manuscript (Wang et al., Curr Med Sci 2022) for incorporation into our manuscript?

Reviewer 2 Report

Comments and Suggestions for Authors

Dear Authors

This article reports the possible mechanisms that interplay with MPO-deficiency and the immune response during inflammation in a murine model of colitis, which is an important topic. However, some concerns in this article need to be addressed.

1.      Please rewrite the abstract section.

2.      Please change “H2O2” to “H2O2”.

3.      Please draw the study design for better understanding.

4.      What was the age of the included mice in the study?

5.      For how long have the mice been acclimatized?

6.      Was DSS administered through the water or via gavage to induce colitis?

7.      Please mention the groups in the method section.

8.      Please specify the number of mice per cage and group.

9.      Were any mice excluded from the study or were there specific exclusion criteria? Please provide clarification and incorporate it into the methods section.

10.  In line 98, please correct the C in the figure 1 legend.

11.     Please explain the abbreviations such as groups under the figures.

12.     Please mention the type and number of mice in the legend of Figures.

13.  Please italicize all the names of the bacteria.

14.  Please include the strengths of your study at the end of discussion section.

Author Response

Reviewer 2

Dear Authors

This article reports the possible mechanisms that interplay with MPO-deficiency and the immune response during inflammation in a murine model of colitis, which is an important topic. However, some concerns in this article need to be addressed.

  1. Please rewrite the abstract section.

We have made some changes to our abstract, in line with Reviewer 1 comments. However, without specific direction, we are unsure with an approach to rewrite the abstract. Could you provide some further guidance here with concerns regarding our abstract?

  1. Please change “H2O2” to “H2O2”.

Thank you for identifying this inconsistency. This was also pointed out by Reviewer 1 and we have since amended our manuscript to all instances of ‘H2O2’

  1. Please draw the study design for better understanding.

We have now incorporated a schematic that illustrates our study design for an easier read and understanding of our method. (line 358)

  1. What was the age of the included mice in the study?

Experiments commenced at 8 weeks of age. Please see excerpt in response to point 6

  1. For how long have the mice been acclimatized?

We have included the sentence in the manuscript below.

‘Mice were acclimatized for 1 week prior to the commencement of experiments’ (line 291)

  1. Was DSS administered through the water or via gavage to induce colitis?

Our DSS was administer orally and we have now amended our methods section to reflect this.

‘Female MPO gene knockout (MPO KO) C57BL/6J mice and corresponding C57BL/6J wild type (Wt) mice were obtained from Australian BioResources . Female mice were chosen in this study as male mice develop more significant and aggressive disease during DSS-induce colitis (10.1002/0471142735.im1525s104). Mice were aged-matched (7 weeks on arrival) and weight-matched (16-17g ± 0.5). Subsequently, mice were housed in the Charles Perkins Centre Animal House (University of Sydney, Australia) in accordance with institutional animal ethics guidelines (Approvals #1288 and #1636) and with ad libitum access to food and water. Mice were acclimatized for 1 week prior to the com-mencement of experiments. Mice were then randomly assigned to their genotype-specific counterparts (wild-type mice, Wt; MPO deficient mice, MPO-/-) and received either water (CTRL) or DSS (DSS) in their drinking water.’

  1. Please mention the groups in the method section.

Please see excerpt in response to point 6.

  1. Please specify the number of mice per cage and group.

The n values are now recorded in the legends of each figure.

  1. Were any mice excluded from the study or were there specific exclusion criteria? Please provide clarification and incorporate it into the methods section.

No mice were excluded from our study as they were ordered with sex, weight and aged-matched as a criteria. This is now detailed in our methods (see response to point 6).

  1. In line 98, please correct the C in the figure 1 legend.

Thank you for pointing out this error. We have now corrected this in our manuscript (line 102-114).

  1. Please explain the abbreviations such as groups under the figures.

The groups have now been explained in the methods section (see response to point 6) to avoid repetition.

  1. Please mention the type and number of mice in the legend of Figures.

We have mentioned the type of mice now in the methods section (see response to point 6) and included the n-value of mice in each respective figure legend (see response to point 8)

  1. Please italicize all the names of the bacteria.

It is an international standard to italicisie all bacteria names and we have now done so in our manuscript.

  1. Please include the strengths of your study at the end of discussion section.

We have now included the strength of our study in the paragraph below

‘The strength of our study is the evaluation of the microbiome at two time-points dur-ing chronic colitis. This has revealed distinct phylogenetic polarities that are largely maintained with time and differences in the relative abundance of some IBD-associated genera in Wt and MPO KO mice which may influence the course of colitis.’ (page 9, line 318-321)